# Curcumin Offers No Additional Benefit to Lifestyle Intervention on Cardiometabolic Status in Patients with Non-Alcoholic Fatty Liver Disease

**DOI:** 10.3390/nu14153224

**Published:** 2022-08-06

**Authors:** Kaveh Naseri, Saeede Saadati, Zahra Yari, Behzad Askari, Davood Mafi, Pooria Hoseinian, Omid Asbaghi, Azita Hekmatdoost, Barbora de Courten

**Affiliations:** 1Gastroenterology and Liver Diseases Research Center, Research Institute for Gastroenterology and Liver Diseases, Shahid Beheshti University of Medical Sciences, Tehran 1985717413, Iran; 2Department of Medicine, School of Clinical Sciences, Monash University, Melbourne, VIC 3168, Australia; 3Department of Nutrition Research, National Nutrition and Food Technology Research Institute, Faculty of Nutrition and Food Technology, Shahid Beheshti University of Medical Sciences, Tehran 1985717413, Iran; 4Faculty of Medicine, Shahid Beheshti University of Medical Sciences, Tehran 1985717413, Iran; 5Cancer Research Center, Shahid Beheshti University of Medical Sciences, Tehran 1985717413, Iran; 6Department of Clinical Nutrition and Dietetics, National Nutrition and Food Technology Research Institute, Shahid Beheshti University of Medical Sciences, Tehran 1985717413, Iran; 7School of Health and Biomedical Sciences, RMIT University, Bundoora, VIC 3083, Australia

**Keywords:** atherogenicity, curcumin, cardiometabolic, NAFLD, insulin resistance, cardiovascular risk factors, nutrition, nutritional supplements

## Abstract

Cardiovascular disease (CVD) is the leading cause of death in patients with non-alcoholic fatty liver disease (NAFLD). Curcumin has been shown to exert glucose-lowering and anti-atherosclerotic effects in type 2 diabetes. Hence, we investigated curcumin’s effects on atherogenesis markers, fatty liver, insulin resistance, and adipose tissue-related indicators in patients with NAFLD. In this secondary analysis of a 12-week randomized controlled trial, fifty-two patients with NAFLD received lifestyle modification. In addition, they were randomly allocated to either the curcumin group (1.5 g/day) or the matching placebo. Outcome variables (assessed before and after the study) were: the fatty liver index (FLI), hepatic steatosis index (HSI), fatty liver score (FLS), BMI, age, ALT, TG score (BAAT), triglyceride glucose (TyG) index, Castelli risk index-I (CRI-I), Castelli risk index-II (CRI-II), TG/HDL–C ratio, atherogenic coefficient (AC), atherogenic index of plasma (AIP), lipoprotein combine index (LCI), cholesterol index (CHOLINDEX), lipid accumulation product (LAP), body adiposity index (BAI), visceral adiposity index (VAI), metabolic score for visceral fat (METS-VF), visceral adipose tissue (VAT), and waist-to-height ratio (WHtR) values. The TyG index decreased in the curcumin group and increased in the placebo group, with a significant difference between the groups (*p* = 0.029). However, a between-group change was not significant after adjustment for multiple testing. Other indices were not significantly different between the groups either before or after multiple test correction. After the intervention, there was a lower number of patients with severe fatty liver (FLI ≥ 60) and metabolic syndrome in the curcumin group compared to the placebo (*p* = 0.021 and *p* = 0.012, respectively). In conclusion, curcumin offers no additional cardiometabolic benefits to lifestyle intervention in patients with NAFLD.

## 1. Introduction

Non-alcoholic fatty liver disease (NAFLD) is the most common cause of chronic liver disease, with a global prevalence of 25.24% [1], which continues to rise with an increasing prevalence of obesity [2]. The worldwide prevalence increased from 15% in 2005 to 25% in 2010 [3], and is more significant in Asian and Pacific countries [4]. NAFLD covers a spectrum of conditions resulting from excessive hepatic fat accumulation (triglycerides), which ranges from steatosis to steatohepatitis, cirrhosis, and fibrosis, and can eventually lead to hepatocellular carcinoma (HCC) [5,6]. A predominant type of NAFLD is non-alcoholic steatohepatitis (NASH), which is the hepatic manifestation of metabolic syndrome that results from an interplay of adipokines released from excess visceral adipose tissue (VAT) and inflammatory cytokines secreted from the macrophages residing in VAT, leading to chronic inflammation and insulin resistance [7,8,9]. Hepatic steatosis is the earliest abnormality in the pathogenesis of NAFLD [10,11]. Importantly, NAFLD is associated with chronic diseases such as type 2 diabetes mellitus (T2DM), metabolic syndrome, obesity, and cardiovascular diseases (CVDs) [12,13], which increases morbidity and mortality from NAFLD [13,14,15,16]. This is primarily due to diets rich in simple sugars and saturated fat, and a sedentary lifestyle, which result in an overload of lipids, mainly triacylglycerol, ectopically stored in the liver. The causal role of inflammation and oxidative stress in NASH is known, though the exact mechanisms are still being elucidated [17].

In addition, lifestyle modification is the main therapeutic intervention in NAFLD. Only a few other therapies, such as metformin, are available for patients with NAFLD, which address both NAFLD and cardiometabolic risk factors [18]. The active component in turmeric, curcumin, has been suggested to possess anti-inflammatory, anti-atherosclerotic, and antioxidative properties that may improve both NAFLD and cardiometabolic conditions. Curcumin has been shown to reduce inflammation markers and oxidative stress, inhibit platelet aggregation, and improve cholesterol homeostasis [19,20]. Moreover, a series of studies have suggested that curcumin is beneficial in decreasing low-density lipoprotein cholesterol (LDL-C) and raising high-density lipoprotein cholesterol (HDL-C), while lowering lipid peroxidation [21,22]. Curcumin has been shown to alleviate atherosclerotic lesions in HCD-induced atherosclerotic rabbits [23].

Emerging evidence suggests that fatty liver indices might be associated with indices of atherosclerosis [24,25,26]. The visceral adiposity index (VAI) and metabolic score for visceral fat (METS-VF) are novel estimates of intra-abdominal fat content and cardio-metabolic health [27,28]. In addition, there is a validated surrogate of body composition-derived adiposity called the body adiposity index (BAI) [29]. Waist-to-height ratio (WHtR) is another indicator closely related to the occurrence of NAFLD [30,31,32,33]. It has also been previously shown that atherogenic indices, such as triglyceride (TG)/HDL-C ratio, Castelli risk index-I (CRI-I), Castelli risk index-II (CRI-II), and triglyceride glucose (TyG) index, can be used as novel monitoring indicators of atherosclerosis and coronary heart disease [34,35,36]. Wang et al. reported that the atherogenic index of plasma (AIP) was strongly correlated with NAFLD among obese individuals, making it clinically useful for NAFLD monitoring [16]. The fatty liver score (FLS), hepatic steatosis index (HSI), fatty liver index (FLI), and BMI, age, ALT, and TG score (BAAT) are markers for the assessment of the severity of NAFLD. Bedogni et al. demonstrated that lipid accumulation product (LAP) was a predictor of ultrasonographic hepatic steatosis among the adult population [37]. Moreover, HSI, FLI, FLS, and TyG have been used in the European cohort and ATTICA study [38,39].

Since inflammation and oxidative stress are important risk factors for NAFLD, metabolic syndrome, and cardiovascular diseases, we proposed that therapies that affect inflammation and oxidative stress, such as curcumin, can provide substantial benefits in tackling several chronic diseases simultaneously. Hence, we investigated the addition of curcumin to lifestyle modification in patients with NAFLD on markers of atherogenicity, fatty liver-associated indices, and adipose tissue-related indicators, such as FLI, FLS, HSI, BAAT, TG to HDL-C ratio, CRI-I, CRI-II, atherogenic coefficient (AC), TyG index, AIP, cholesterol index (CHOLINDEX), lipoprotein combine index (LCI), LAP, BAI, VAI, METS-VF, VAT, and WHtR.

## 2. Materials and Methods

### 2.1. Study Design

The study design, methodology, consolidated standards of reporting trials diagram, and preliminary results of the double-blind, placebo-controlled, randomized clinical trial have been previously published [40,41,42]. The current trial was confirmed by the Ethics Committee of the National Nutrition and Food Technology Research Institute at Shahid Beheshti University of Medical Sciences (IR.SBMU.NNFTRI.1395.106) and carried out in accordance with the Declaration of Helsinki guidelines. All subjects signed the designed written informed consent before enrollment in the trial. The present study is a post-hoc analysis of the placebo-controlled trial, which evaluated the effect of curcumin intake plus lifestyle modification on atherogenicity, fatty liver and indicators of insulin resistance, and adipose tissue in patients with NAFLD.

### 2.2. Study Participants and Intervention

Subjects aged 18 years or older diagnosed with NAFLD (CAP > 263 dB/m in FibroScan) were selected for the study. Any subject with a history of diabetes, biliary disorders, cirrhosis, hepatitis, malignancies, autoimmune disorders, hypothyroidism, hypertension, Cushing’s syndrome, respiratory, cardiovascular, renal disorders, and alcohol use was excluded. Furthermore, subjects who used metformin, methotrexate, ursodeoxycholic acid, vitamin E, corticosteroids, phenytoin, lithium, and tamoxifen within 12 weeks of the study or underwent bariatric surgery or more than 10% weight loss from baseline body weight during the intervention period for any reason, pregnancy, unexpected adverse effects, and supplement intolerance were removed from the trial.

Eligible subjects were block randomized based on gender and body mass index (BMI), and were assigned to either the curcumin or control group. The randomization sequence was generated using a computer by an independent statistician with an equal allocation ratio of 1:1. Participants and the project manager (dietitian) were totally blind to the intervention and control groups. In addition to the dietary recommendations that have already been described in detail, all subjects were advised to exercise 30 min thrice a week [40]. The intervention group received 500 mg curcumin capsules (Arjuna Natural Extract, India) thrice daily, and the control group was given 500 mg placebo capsules thrice daily for 12 weeks.

Turmeric rhizomes identified taxonomically as Curcuma longa were dried, extracted in ethyl acetate, and distilled with water to obtain a turmeric extract. Each 500 mg turmeric extract capsule contained 475 mg of curcuminoids and essential oil (turmerones). Placebo capsules were filled with maltodextrin, identically matching the turmeric extract capsules in color, shape, and size. To conceal the allocation, turmeric and placebo capsules were filled in sequentially numbered identical pre-randomized bottles labeled as A or B. An independent pharmacist dispensed the study medication to participants and maintained drug accountability. Compliance with the intervention was assessed at each visit by recording the daily consumption of capsules. Those who did not consume at least 90 percent of the scheduled capsules were excluded. All participants were asked to modify their diet, and an energy and nutrient-balanced diet plan was recommended. A balanced diet and exercise were ensured via phone and face-to-face visits. At the end of the trial, the investigator provided the randomization code in sealed opaque envelopes that contained encoding and information about the intervention.

### 2.3. Data Collection

Anthropometric parameters were measured for participating patients at the baseline and the end of the trial. Results from the laboratory analysis have been reported previously [40,41,42]. All participants underwent hepatic fibrosis and steatosis assessment at the beginning and end of the study, applying FibroScan (Echosense, France). Detailed descriptions of the procedures used have been reported previously [40,41,42].

### 2.4. Definition of the Fatty Liver, Atherosclerosis, and Adipose Tissue-Related Indices

Simple, cost-effective, and non-invasive tests for NAFLD are preferred over liver biopsy, and LAP is a valid alternative [43]. LAP performs better than BMI as a marker for NAFLD and is also a risk factor for diabetes [44], metabolic syndrome [45,46], and cardiovascular diseases [47,48]. In addition, the TyG index is a novel marker with high specificity and sensitivity in diagnosing metabolic syndrome [49]. Additionally, the best-validated steatosis biomarker-related index is the FLI, which is also helpful for evaluating NAFLD, with a high correlation with imaging and histological findings [43], as well as being beneficial for assessing CVD, since the risk factors are the same for both [43]. Furthermore, adipose tissue-related indicators have been correlated with higher morbidity and mortality rate among patients with NAFLD [27,30]. Diabetes mellitus, alanine transaminase (ALT)/aspartate transaminase (AST) ratio, sex, and BMI were independently predictive of NAFLD and are used to calculate the HSI. Shreds of evidence support that the FLI seems inappropriate in assessing NAFLD in a population whose BMI and waist circumference (WC) are substantially lower and, in such cases, HSI is more reliable.

Evaluation of the fatty liver, insulin resistance, atherosclerotic-related indices, and adipose tissue indicators was done using the following formulas in the current post-hoc randomized controlled trial (RCT) analysis [50,51]:(1)$$\mathrm{CRI-I}=\frac{total\;cholestrol\;\left(\frac{mmol}{L}\right)}{HDL-C\;\left(\frac{mmol}{L}\right)}$$

[50,51]
(2)$$\mathrm{CRI-II}=\frac{LDL-C\;\left(\frac{mmol}{L}\right)}{HDL-C\;\left(\frac{mmol}{L}\right)}$$

[52]
(3)$$\mathrm{TG}\;\mathrm{to}\;\mathrm{HDL-C}\;\mathrm{ratio}=\frac{triglycerides\;\left(\frac{mmol}{L}\right)}{HDL-C\;\left(\frac{mmol}{L}\right)}$$

[52]
(4)$$\mathrm{TyG}\;\mathrm{index}=Ln\;\left(\frac{fasting\;triglycerides\;\left(\frac{mg}{dL}\right)\times fasting\;glucose\;\left(\frac{mg}{dL}\right)}{2}\right)$$

[51]
(5)$$\mathrm{AC}=\frac{total\;cholestrol\;\left(\frac{mmol}{L}\right)-HDL-C\;\left(\frac{mmol}{L}\right)}{HDL-C\;\left(\frac{mmol}{L}\right)}$$

[50,51]
(6)$$\mathrm{AIP}=Log_{10}\;\left(\frac{triglycerides\;\left(\frac{mmol}{L}\right)}{HDL-C\;\left(\frac{mmol}{L}\right)}\right)$$

[50,53]
(7)$$\mathrm{LCI}=\frac{total\;cholestrol\;\left(\frac{mmol}{L}\right)\times triglycerides\;\left(\frac{mmol}{L}\right)\times LDL-C\;\left(\frac{mmol}{L}\right)}{HDL-C\;\left(\frac{mmol}{L}\right)}$$

[54,55]
(8)$$\mathrm{CHOLINDEX}=\frac{LDL-C\;\left(\frac{mmol}{L}\right)-LDL-C\;\left(\frac{mmol}{L}\right)}{HDL-C\;\left(\frac{mmol}{L}\right)}$$

LAP =

(WC (cm) − 65) × (TG concentration (mmol/L)) for males

(WC (cm) − 58) × (TG concentration (mmol/L)) for females

[46]



(9)
$$\mathrm{FLI}=\frac{\left(e^{0.953\times \mathrm{loge}(\mathrm{triglycerides})+0.139\times \mathrm{BMI}+0.718\times \mathrm{loge}(\mathrm{GGT})+0.053\times \mathrm{waistcircumference}-15.745}\right)}{\left(1+e^{0.953\times \mathrm{loge}(\mathrm{triglycerides})+0.139\times \mathrm{BMI}+0.718\times \mathrm{loge}(\mathrm{GGT})+0.053\times \mathrm{waistcircumference}-15.745}\right)}\times 100$$



[56]
HSI = 8 × (ALT/AST ratio) + BMI (+2, if female; +2, if diabetes mellitus)(10)

[57]
FLS = 1.18 × metabolic syndrome + 0.45 × diabetes (2, if yes; 0, if no) + 0.15 × FSI (mU/L) + 0.04 × AST (U/L) − 0.94 × (AST/ALT) − 2.89 (11)

[58]

BAAT = was calculated as the sum of the following categorical variables:BMI (≥28 = 1, <28 = 0), age at liver biopsy (≥50 years =; <50 = 0), ALT (≥2N = 1, <2N = 0), and serum triglycerides (≥1.7 mmol/l = 1, <1.7 = 0), thus ranging from 0 to 4.(12)

[59]

(All patients had TG < 400 mg/dL)
BAI = hip circumference (cm)/height (m)1.5–18(13)

[29]
(14)$$\mathrm{VAI}=\left(\frac{WC}{36.58+\left(1.89\times BMI\right)}\right)\times \left(\frac{TG}{0.81}\right)\times \left(\frac{1.52}{HDL}\right)\;\mathrm{for}\;\mathrm{females}$$
(15)$$\left(\frac{WC}{39.68+\left(1.88\times BMI\right)}\right)\times \left(\frac{TG}{1.03}\right)\times \left(\frac{1.31}{HDL}\right)\;\mathrm{for}\;\mathrm{males}$$

(The WC, BMI, TG, and HDL values are expressed in cm, kg/m^2^, and mol/L, respectively.)

[60]
METS-VF = 4.466 + 0.011 × (Ln (METS-IR))^3^ + 3.239 × (Ln (WHtr))^3^ + 0.319 × (Sex) +(16)

0.594 × (Ln (Age))

[27]
VAT volume (cm^3^) = 47.03 × age (yr) + 117.79 × BMI (kg/m^2^) + 74.18 × WC (cm) − 8792.7(17)

[30]

WHtR = WC/height

[33]

All recommendations and measurements were given/performed by the same researcher to avoid intra-observer biases. 

### 2.5. Statistical Analysis 

The descriptive statistics were provided for quantitative and categorical variables, representing mean change and standard deviation (SD) or frequencies (*n*). Following the testing for normality by applying the Kolmogorov–Smirnov statistic, an independent sample t-test was applied to analyze between-group alterations of quantitative variables. In addition, a McNemar test was run for categorical data. Within-group differences during the trial were also compared using a paired sample t-test. To investigate the impacts of curcumin compared to the placebo on atherogenicity and fatty liver-associated markers, and adipose tissue-related indices among the participants, an analysis of covariance (ANCOVA) was conducted which integrated BMI levels, baseline values, and energy intake. Relatedly, means (adjusted) and 95% confidence intervals (95% CI) were provided. Post-hoc analysis was performed using the Bonferroni–Dunn method to adjust for multiple comparisons [61]. The statistical analyses were performed using SPSS Inc., Chicago, USA, version 24, and Graphpad PRISM 8.0 software (Graphpad Software, San Diego, CA, USA). For the statistical significance of the performed tests, a *p*-value < 0.05 was considered.

## 3. Results

### 3.1. Participants’ Recruitment Procedure and Baseline Characteristics

The studied subjects’ recruitment procedure and detailed results of the serum metabolic parameters have been described elsewhere [40,42]. The participation rate in the present study was 96%. Briefly, 27 patients were treated with curcumin (1.5 g/day), and 23 were treated with a placebo for 12 weeks. Participants’ clinical and demographic characteristics were homogeneously distributed between the two groups (*p* > 0.05).

### 3.2. Changes in Fatty liver-Associated Indicators

The changes in fatty liver-associated indicators of participants before and after consumption of curcumin or the placebo, plus a lifestyle intervention in a 12-week RCT, are shown in Table 1. At the beginning of the study, no differences were seen between the studied groups. After 12 weeks of follow-up, all indices tended to decrease in both groups. Lifestyle intervention resulted in a decrease in FLI and FLS in both treatment groups (FLI: *p* = 0.002 for curcumin and *p* = 0.01 for the placebo; FLS: *p* ˂ 0.001 for curcumin and *p* = 0.021 for the placebo, respectively). Within-group comparisons demonstrated a reduction in BAAT (*p* = 0.076) in the curcumin group, which was approaching significance. However, there was no significant difference between the two groups. In addition, after controlling for multiple testing, no significant between-group change was replicated regarding the fatty liver-associated indices. It is also worth noting that adjustment for confounding factors did not change the significance of the differences.

### 3.3. Changes in Atherogenicity and Insulin Resistance Markers

The mean ± SD of atherogenic indices at the beginning and at the end of the trial and their changes during the trial are shown in Table 2. No significant changes were noted in the baseline distribution of atherogenic indices between the patients in curcumin and placebo groups. There was a difference in the TyG index between the two groups (−0.16 vs. 0.09, *p* = 0.029). However, after controlling for multiple testing, all the atherogenecity markers, including the TyG index, showed a non-significant change between the curcumin and control groups. After adjustment for the baseline values, body mass index, and energy intake, the between-group difference was no longer significant (*p* = 0.056). Other atherogenic indices, however, did not show significant differences between the two groups.

### 3.4. Changes in Adipose Tissue-Related Indicators

Table 3 summarizes the changes in adipose tissue-related indicators among individuals with NAFLD at baseline and after intervention in both the intervention and placebo groups. The baseline levels of LAP, BAI, VAI, METS-VF, VAT, and WHtR did not show any differences between the curcumin or placebo groups (*p* > 0.05). Within-group comparisons demonstrated a significant reduction in LAP (*p* = 0.025), BAI (*p* = 0.01), METS-VF (*p* < 0.001), VAT (*p* = 0.001), and WHtR (*p* < 0.001) in the curcumin group. Furthermore, BAI (*p* = 0.02), METS-VF (*p* = 0.003), VAT (*p* = 0.004), and WHtR (*p* = 0.02) were also significantly reduced in the placebo group. After adjusting for confounders and multiple testing, no statistically significant differences were observed between the groups (all *p* > 0.05).

### 3.5. Changes in the Frequency of Comorbidities

Metabolic syndrome, diabetes (FPG ≥ 100 mg/dL), and fatty liver (based on FLI ≥ 60, HSI ≥ 36, FLS > -0.64, BAAT ≥ 2) are presented as frequency. The difference at baseline and 16-week follow-up was analyzed using the McNemar test (Figure 1a–c). The frequency of patients with fatty liver and metabolic syndrome at the end of the intervention was significantly lower in the curcumin group than in the control group. 

## 4. Discussion

We have shown that adding 1.5 g of curcumin daily to a lifestyle modification for 12 weeks weeks had no beneficial effect on cardiometabolic health in patients with NAFLD. After the intervention, there was a lower number of patients with severe fatty liver (FLI ≥ 60) and metabolic syndrome in the curcumin group compared to the placebo. To the best of our knowledge, this is the first controlled trial to explore the effects of curcumin, plus lifestyle intervention, on hepatic health indices, atherogenic markers, and adipose tissue-related indicators, instead of using conventional lipid components. These markers represent a better balance between plasma lipoproteins and can be used to monitor the treatment response in research and clinical settings better than the conventional single lipid parameter.

Patients with NAFLD suffer from an increased risk of cardiovascular morbidity and mortality [62]. At present, there are no specific food and drug administration (FDA)-approved drugs for NAFLD. The treatment is largely based on lifestyle optimization. Nevertheless, several nutraceuticals have been shown to reduce lipid infiltration of the liver, as well as improve anthropometric, haemodynamic, and cardiometabolic outcomes [63].

Lifestyle modifications, including reduced caloric intake, and physical activity, are the most effective non-pharmacological approaches for improving NAFLD [64]. Consistent with this, we found that lifestyle intervention decreased steatosis scores, specifically FLI and FLS, and adipose tissue-related indicators, including BAI, METS-VF, VAT, and WHtR, in both treatment groups. In a trial by Nourian et al., health belief model (HBM)-based lifestyle intervention education was effective in improving liver function tests and fatty liver grade assessed by ultra-sonography of NAFLD patients [65]. In another study using different exercise modalities, low to moderate-intensity exercise produced a significant but small decrease in liver fat, regardless of the type of exercise [66]. In a 4-week exercise program of aerobic cycling, the levels of hepatic triglycerides were reduced, while Homeostatic Model Assessment for Insulin Resistance (HOMA-IR) was not affected [67]. Additionally, a 3-month hypocaloric diet plus aerobic exercise of 60 min 3× per week resulted in significant reductions in ALT, AST, body weight, and BMI values [68]. Among a cohort of non-diabetic obese individuals, lifestyle modification resulted in a more significant reduction in VAT in those with NAFLD compared to subjects without NAFLD [31]. It has been previously reported that the level of physical activity, as a major component of lifestyle modification, is correlated with WHtR values among individuals with NAFLD in a cross-sectional setting [32].

The TyG index has been suggested as an indicator of atherosclerosis and metabolic syndrome, which seems to be associated with glucotoxicity, lipotoxicity, cardiometabolic, including obesity, type 2 diabetes, fatty liver, hypertension, CAD, and arterial stiffness [49,52,69,70]. Particularly, the TyG index uses serum fasting TG levels and fasting plasma glucose (FPG), and is a surrogate marker of insulin resistance. This could be beneficial, considering that HOMA-IR is expensive due to the insulin measurement [52,71,72]. In the present study, we found a decrease in the TyG index after a 12-week treatment with 1.5 g curcumin. We performed multiple testing to avoid a type 1 error [73]. However, after controlling for multiple testing, the significant between-group difference disappeared. In addition, we did not see any change in HOMA-IR in the curcumin vs. placebo group in our study [40]. Therefore, the null effect of curcumin on TyG and HOMA-IR indicated that curcumin had no effect on insulin resistance. Curcumin has been shown to enhance glycolysis and glycogen synthesis, and reduce gluconeogenesis in the liver, and increase glycogen synthesis, glycolysis, and glucose uptake in skeletal muscles, resulting in an improved insulin release and better glycemic control [74]. However, the effectiveness of curcumin on insulin resistance in the literature is inconsistent [75,76]. We did not find a significant improvement in the TyG index at the end of the study; the TyG index decreased in the curcumin group and increased in the placebo group. Curcumin supplementation reduced the number of patients with features of metabolic syndrome by 60%, compared with only 14% in the control group.

We have previously shown that consuming 1.5 g curcumin in addition to lifestyle intervention did not change hepatic steatosis and fibrosis, lipid profile, glucose, insulin resistance, ALT, anthropometric indices, [40] tumor necrosis-α (TNF-α), and nuclear factor-kappa B (NF-*k*B) activity [42]. The current study’s results are in concordance with some trials, while there are other investigations with positive changes in lipid profiles [77], liver enzymes, and anthropometric indices [78,79,80]. In agreement with this, the meta-analysis on nine RCTs found that consuming curcumin improves dyslipidemia, specifically reducing serum total cholesterol (TC) and LDL-C, ALT, AST, FPG, HOMA-IR, serum insulin, and WC, but not in serum TG, HDL-C, hemoglobin A1c (HbA1c), body weight, and BMI [81]. Additionally, the anti-inflammatory and anti-oxidative properties of curcumin in various health conditions, through its lowering effects on C-reactive protein, interleukin-6, and malondialdehyde, have been demonstrated [82]. However, we did not show an effect of curcumin on inflammation, lipid profile, and most of the atherogenic indices in this study. Normal lipid profile values at the beginning of the study, significant weight loss in both intervention and placebo groups, short intervention duration, and a low dose of curcumin may be the reasons for the lack of effect in this study.

However, reductions in most of the atherogenic indices were observed in the present study and none were statistically significant, suggesting that lifestyle intervention is sufficient to change these patients’ cardiovascular risk. In addition, there were no significant differences in the fatty liver-associated indices values compared with the control group. The lack of effect of curcumin on lipid profile, which has been shown in various studies, was also confirmed in the current study. Normal levels of lipid profile at the beginning of the study, short intervention duration, and a low dose of curcumin are some of the reasons that can explain the lack of significant change in atherogenic indices in this study. Notably, curcumin supplementation, in addition to lifestyle modification, did not change adipose tissue-related indicators compared to the placebo group in the present study. This finding is in line with previous preclinical evidence [83]. This suggests that curcumin has no favorable effects on excessive visceral adipose tissue in patients with NAFLD. However, more preclinical and clinical trials are warranted to confirm this finding.

Our study has some strengths: as far as we know, no similar clinical trials using inexpensive, simple means for assessing CVD risk have been conducted; rigorous randomized controlled placebo-controlled design of the study, adding lifestyle modification to the curcumin supplementation; and the inclusion of newly diagnosed NAFLD patients who had not received any therapeutic intervention. Although the present results appear promising, this study had some limitations. Firstly, a short length of follow-up leads to the inability to assess the long-term efficacy of curcumin consumption on atherogenicity, fatty liver-associated indices, and adipose tissue-related indicators, as well as improvement of metabolic complications. Secondly, the small sample size introduces other potential limitations.

## 5. Conclusions

Taken together, compared to lifestyle modification alone, which can be considered as a cornerstone in reducing liver fat content and visceral adiposity, curcumin offers no additional cardiometabolic benefits to patients with NAFLD. It is also worth noting that more extensive trials with longer follow-ups are needed to confirm our findings.

## Figures and Tables

**Figure 1 nutrients-14-03224-f001:**
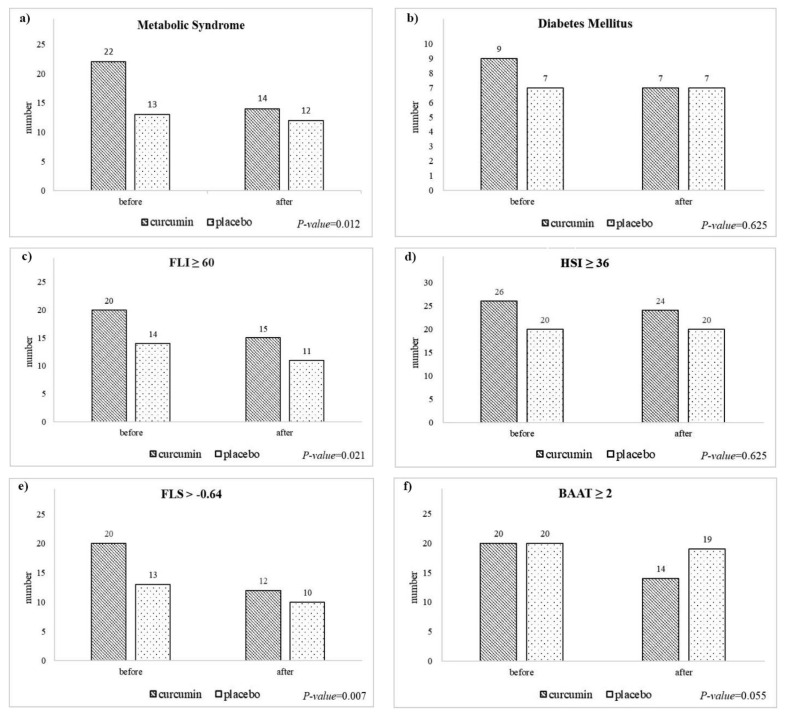
The frequency of patients with fatty liver, diabetes mellitus, and metabolic syndrome. (**a**) metabolic syndrome; (**b**) diabetes mellitus; (**c**) fatty liver index (FLI); (**d**) hepatic steatosis index (HSI); (**e**) fatty liver score (FLS); (**f**) BMI, age, ALT, TG score (BAAT).

**Table 1 nutrients-14-03224-t001:** Changes in LAP and fatty liver-associated indicators.

	Curcumin Group (*n* = 27)	Placebo Group (*n* = 23)	*p*-Value *	Adjusted *p*-Value ^¥^
Fatty Liver Index (FLI)				
Baseline	74.87 ± 17.69	74.62 ± 20.87	0.9	>0.999
After 12 weeks	63.51 ± 26.39	66.28 ± 28.24	0.7	>0.999
Differences 95% CI	−11.36 (−18.2, −4.4)	−8.3 (−14.4, −2.2)	0.5	>0.999
*p*-value ^⁑^	0.002	0.01	0.7	
Hepatic Steatosis Index (HSI)				
Baseline	47.49 ± 7.84	49.37 ± 8.78	0.4	>0.999
After 12 weeks	45.06 ± 9.05	49.35 ± 17.28	0.3	0.9
Differences 95% CI	−2.42 (−7.19, 2.3)	−0.1 (−8.9, 8.9)	0.6	0.2
*p*-value ^⁑^	0.3	0.9	0.4 ^⁋^	
Fatty Liver Score (FLS)				
Baseline	0.19 ± 1.28	0.65 ± 1.66	0.3	0.9
After 12 weeks	−0.68 ± 1.3	−0.12 ± 1.6	0.2	0.6
Differences 95% CI	−0.87 (−1.3, −0.5)	−0.78 (−1.4, −0.13)	0.8	>0.999
*p*-value ^⁑^	<0.001	0.021	0.7 ^⁋^	
BMI, Age, ALT, TG score (BAAT)				
Baseline	2 ± 0.73	1.47 ± 1.12	0.065	0.2
After 12 weeks	1.63 ± 1.08	1.47 ± 0.94	0.6	>0.999
Differences 95% CI	−0.37 (−0.78, 0.04)	0 (−0.5, 0.5)	0.2	0.7
*p*-value ^⁑^	0.07	1	0.9 ^⁋^	

* Independent *t*-test. ^⁑^ Paired *t*-test. ^⁋^ ANCOVA, adjusted for baseline values, changes in MET, and energy intake. ^¥^ Correction for multiple comparisons using the Bonferroni–Dunn method.

**Table 2 nutrients-14-03224-t002:** Comparison of atherogenic indicators before and after of the trial.

	Curcumin Group (*n* = 27)	Placebo Group (*n* = 23)	*p*-Value *	Adjusted *p*-Value ^¥^
Castelli Risk Index-I (CRI-I)				
Baseline	4.59 ± 0.75	4.80 ± 0.82	0.4	>0.999
After 12 weeks	4.62 ± 1.57	4.69 ± 0.95	0.8	>0.999
Differences 95% CI	0.02 (−0.6, 0.6)	−0.11 (−0.47, 0.25)	0.7	>0.999
*p*-value ^⁑^	0.9	0.5	0.8 ^⁋^	
Castelli Risk Index-II (CRI-II)				
Baseline	2.77 ± 0.72	3.06 ± 0.85	0.2	0.7
After 12 weeks	2.78 ± 0.96	2.86 ± 0.91	0.8	>0.999
Differences 95% CI	0.01 (−0.38, 0.4)	−0.19 (−0.5, 0.1)	0.4	>0.999
*p*-value ^⁑^	0.9	0.2	0.4 ^⁋^	
TG/HDL-C				
Baseline	1.79 ± 0.81	1.62 ± 0.84	0.5	>0.999
After 12 weeks	1.90 ± 1.75	1.81 ± 0.89	0.8	>0.999
Differences 95% CI	0.11 (−0.57, 0.79)	0.19 (−0.07, 0.4)	0.8	>0.999
*p*-value ^⁑^	0.7	0.1	0.8 ^⁋^	
TyG index				
Baseline	8.9 ± 0.39	8.78 ± 0.55	0.4	>0.999
After 12 weeks	8.73 ± 0.44	8.87 ± 0.53	0.3	>0.999
Differences 95% CI	−0.16 (−0.35, 0.02)	0.09 (−0.05, 0.23)	0.029	0.1
*p*-value ^⁑^	0.08	0.2	0.056 ^⁋^	
Atherogenic Coefficient (AC)				
Baseline	3.59 ± 0.75	3.8 ± 0.82	0.4	>0.999
After 12 weeks	3.62 ± 1.57	3.69 ± 0.95	0.8	>0.999
Differences 95% CI	0.02 (−0.6, 0.6)	−0.11 (−0.47, 0.25)	0.7	>0.999
*p*-value ^⁑^	0.9	0.5	0.8 ^⁋^	
Atherogenic Index of Plasma (AIP)				
Baseline	0.22 ± 0.17	0.16 ± 0.21	0.3	0.9
After 12 weeks	0.19 ± 0.24	0.21 ± 0.2	0.7	>0.999
Differences 95% CI	−0.02 (−0.11, 0.06)	0.05 (0, 0.1)	0.2	0.6
*p*-value ^⁑^	0.5	0.07	0.3 ^⁋^	
Lipoprotein Combine Index (LCI)				
Baseline	25.51 ± 15.42	26.1 ± 15.19	0.9	>0.999
After 12 weeks	20.95 ± 12.92	27.43 ± 15.51	0.1	0.4
Differences 95% CI	−4.56 (−10.7, 1.6)	1.33 (−5.2, 7.86)	0.2	0.6
*p*-value ^⁑^	0.1	0.6	0.2 ^⁋^	
Cholesterol Index (CHOLINDEX)				
Baseline	1.77 ± 0.72	2.06 ± 0.85	0.2	0.7
After 12 weeks	1.78 ± 0.96	1.86 ± 0.91	0.8	>0.999
Differences 95% CI	0.01 (−0.4, 0.4)	−0.19 (−0.54, 0.15)	0.4	>0.999
*p*-value ^⁑^	0.9	0.2	0.4 ^⁋^	

* Independent *t*-test. ^⁑^ Paired *t*-test. ^⁋^ ANCOVA, adjusted for baseline values, changes in MET, and energy intake. ^¥^ Correction for multiple comparisons using the Bonferroni–Dunn method.

**Table 3 nutrients-14-03224-t003:** Adipose tissue-related indices before and after the study.

	Curcumin Group (*n* = 27)	Placebo Group (*n* = 23)	*p*-Value *	Adjusted *p*-Value ^¥^
Lipid Accumulation Product (LAP)				
Baseline	75.01 ± 37.16	69.80 ± 35.36	0.6	>0.999
After 12 weeks	59.79 ± 33.78	71.27 ± 39.25	0.3	0.9
Differences 95% CI	−15.21 (−28.3, −2)	1.47 (−12.4, 15.3)	0.07	0.2
*p*-value ^⁑^	0.025	0.8	0.09 ^⁋^	
Body adiposity index (BAI)				
Baseline	37.15 ± 7.42	34.52 ± 7.23	0.2	0.8
After 12 weeks	35.53 ± 7.47	32.86 ± 7.51	0.2	0.7
Differences 95% CI	−1.6 (−2.8, −0.4)	−1.5 (−2.89, −0.19)	0.9	0.9
*p*-value ^⁑^	0.011	0.028	0.7 ^⁋^	
visceral adiposity index (VAI)				
Baseline	2.93 ± 1.48	2.49 ± 1.12	0.3	0.9
After 12 weeks	3.14 ± 3.43	2.78 ± 1.22	0.6	>0.999
Differences 95% CI	0.21 (−1.05, 1.48)	0.28 (−0.09, 0.66)	0.9	>0.999
*p*-value ^⁑^	0.7	0.1	0.9 ^⁋^	
METS-VF				
Baseline	7.17 ± 0.35	7.17 ± 0.41	0.9	>0.999
After 12 weeks	7.01 ± 0.48	7 ± 0.56	0.9	>0.999
Differences 95% CI	−0.15 (−0.23, −0.77)	−0.16 (−0.25, −0.06)	0.9	>0.999
*p*-value	<0.001	0.003	0.7 ^⁋^	
Visceral adipose tissue (VAT) (cm^3^)				
Baseline	4764.9 ± 1476	4736.15 ± 1372.3	0.9	>0.999
After 12 weeks	4273.7 ± 1517.9	4382.4 ± 1684.5	0.8	>0.999
Differences 95% CI	−491.2 ± 417.8	−437.2 ± 562.1	0.7	>0.999
*p*-value ^⁑^	<0.001	0.004	0.8 ^⁋^	
Waist to height ratio (WHtR)				
Baseline	0.63 ± 0.07	0.62 ± 0.07	0.5	>0.999
After 12 weeks	0.59 ± 0.07	0.6 ± 0.09	0.9	>0.999
Differences 95% CI	−0.03 (−0.04, −0.02)	−0.02 (−0.04, −0.002)	0.2	0.9
*p*-value ^⁑^	<0.001	0.029	0.3 ^⁋^	

* Independent *t*-test. ^⁑^ Paired *t*-test. ^⁋^ ANCOVA, adjusted for baseline values, changes in MET, and energy intake. ^¥^ Correction for multiple comparisons using the Bonferroni-Dunn method.

## Data Availability

The datasets used and/or analyzed during the current study are available from the corresponding author upon reasonable request.

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
