# Peer review of "Curcumin Offers No Additional Benefit to Lifestyle Intervention on Cardiometabolic Status in Patients with Non-Alcoholic Fatty Liver Disease"

_nutrients, 2022, doi:10.3390/nu14153224_

Round 1

Reviewer 1 Report

This is a sequel to several papers that describe the results of a nicely executed trial in which treatment with curcumin is compared with placebo and lifestyle intervention. Confirming many studies these papers show that lifestyle intervention ammeliorates almost all factors measured (refs40-42 in the present study). Curcumin has very little or no effect. To optimize the analysis the authors now calculated all the diferent metabolic indexes they could find in the literature and almost obviously found that no effect on primary data translates to no effect on indices. There was one exception the TyG index that did show a barely significant improvement after curcumin treatment. However, as far as I could see no correction for multiple testing was carried out making also this conclusion questionable.  

Author Response

Response to Reviewers:

We wish to thank the reviewers for their time and effort in helping us improve our paper.

Reviewer #1:

Comment 1: Moderate English changes required.

Response: We conducted a thorough English revision of the manuscript.

Comment 2: This is a sequel to several papers that describe the results of a nicely executed trial in which treatment with curcumin is compared with placebo and lifestyle intervention. Confirming many studies these papers show that lifestyle intervention ameliorates almost all factors measured (refs 40-42 in the present study). Curcumin has very little or no effect. To optimize the analysis, the authors now calculated all the different metabolic indexes they could find in the literature and almost obviously found that no effect on primary data translates to no effect on indices. There was one exception the TyG index that did show a barely significant improvement after curcumin treatment. However, as far as I could see no correction for multiple testing was carried out making also this conclusion questionable. 

Response: Thank you for your review. The conclusion parts of the text and the abstract were re-written more accurately. We have also controlled for multiple testing.

Reviewer 2 Report

So, the curcumin is not the first-line drug for the treatment of the II-IV stages of NAFLD, but it’s can use with steatosis as a nutritional additive and in other liver diseases as a seasoning for cooking.

Author Response

Response to Reviewers:

We wish to thank the reviewers for their time and effort in helping us improve our paper.

Reviewer #2:

Comment 1: So, the curcumin is not the first-line drug for the treatment of the II-IV stages of NAFLD, but it can be used with steatosis as a nutritional additive and in other liver diseases as a seasoning for cooking.

Response: Thank you for your review.

Reviewer 3 Report

Cardiovascular disease (CVD) is the leading cause of death in patients with non-alcoholic fatty liver disease (NAFLD). Lifestyle modification is the main therapeutic intervention, however only several pharmacological therapies are available for addressing both NAFLD and cardiometabolic risk factors in patients. The present study examined the benefit of curcumin in addition to lifestyle intervention on cardiometabolic status in patients with NAFLD. The study found that TyG index, a marker of cardiometabolic health, is decreased in the curcumin group and increased in the placebo group following a 12-week intervention. After the intervention, there was a lower number of patients with severe fatty liver and metabolic syndrome in the curcumin group compared to placebo. The study concluded that curcumin offers additional cardiometabolic benefits to lifestyle modifications in patients with NAFLD.

Minor comments:

11)      As inflammation and oxidative stress are drivers of both CVD and NAFLD, and that curcumin is known to reduce both of these, would be great if the authors can include a sentence or two in the discussion to shed light on this.

22)      Line 61-62, “…deposited in the liver” suggest to change to “ectopically stored in the liver”.

33)      Line 182-183, the “FLI” equation seems to have formatting error.

44)      Figure 1, the graphs are very faint, y-axis line is not visible, the key for the curcumin/placebo is too small, “metabolic syndrome” should begin with capital letter.

55)      Line 272-273, “better a balance” please delete “a”.

66)      Line 295-296, “the level of physical activity, as a major components of lifestyle modification, was correlated with WHtR values”, sudden increase in font size, please check.

77)      Line 304, please add “curcumin” after 1.5g.

88)      Line 344, did the authors mean “…assessing CVD risk have been conducted”?

Author Response

Response to Reviewers:

We wish to thank the reviewers for their time and effort in helping us improve our paper.

Reviewer #3:

Comment 1: English language and style are fine/minor spell check required.

Response: The English revision was done throughout the manuscript.

Comment 2: As inflammation and oxidative stress are drivers of both CVD and NAFLD, and that curcumin is known to reduce both of these, would be great if the authors can include a sentence or two in the discussion to shed light on this.

Response: The sentences related to this part were added (lines 340-345).

Comment 3: Line 61-62, “…deposited in the liver” suggest to change to “ectopically stored in the liver”.

Response: This was changed to “ectopically stored in the liver” (line 64).

Comment 4: Line 182-183, the “FLI” equation seems to have formatting error.

Response: The error was corrected (line 184).

Comment 5: Figure 1, the graphs are very faint, y-axis line is not visible, the key for the curcumin/placebo is too small, “metabolic syndrome” should begin with capital letter.

Response: The mentioned changes were applied.

Comment 6:  Line 272-273, “better a balance” please delete “a”.

Response: This was deleted (line 286).

Comment 7: Line 295-296, “the level of physical activity, as a major components of lifestyle modification, was correlated with WHtR values”, sudden increase in font size, please check.

Response: The font size was corrected (line 309).

Comment 8: Line 304, please add “curcumin” after 1.5g.

Response: The word “curcumin” was added (line 318).

Comment 9: Line 344, did the authors mean “…assessing CVD risk have been conducted”?

Response: Thank you. The wording was corrected (line 361).

Round 2

Reviewer 1 Report

Auhors now corrected for multiple testing which improved the manuscript